# Response of Vitamin D after Magnesium Intervention in a Postmenopausal Population from the Province of Granada, Spain

**DOI:** 10.3390/nu12082283

**Published:** 2020-07-30

**Authors:** Héctor Vázquez-Lorente, Lourdes Herrera-Quintana, Jorge Molina-López, Yenifer Gamarra-Morales, Beatriz López-González, Claudia Miralles-Adell, Elena Planells

**Affiliations:** 1Department of Physiology, School of Pharmacy, Institute of Nutrition and Food Technology “José Mataix”, University of Granada, 18071 Granada, Spain; lourdesherra@ugr.es (L.H.-Q.); beatrizlogo@yahoo.es (B.L.-G.); cmirallesadell@gmail.com (C.M.-A.); elenamp@ugr.es (E.P.); 2Department of Physical Education and Sports, Faculty of Education, Psychology and Sports Sciences, University of Huelva, 21007 Huelva, Spain; jorge.molina@ddi.uhu.es

**Keywords:** vitamin D, magnesium, liquid chromatography—tandem mass spectrometry, post-menopause

## Abstract

Menopause is a stage of hormonal imbalance in women which, in addition to other physiopathological consequences, poses a risk of deficiency of key micronutrients such as magnesium and vitamin D. A study was made of the influence of a magnesium intervention upon vitamin D status in a postmenopausal population from the province of Granada (Spain). Fifty-two healthy postmenopausal women between 44–76 years of age were included. Two randomized groups—placebo and magnesium (500 mg/day)—were treated during eight weeks. Nutrient intake was assessed using questionnaires based on 72-h recall. Vitamin D was analyzed by liquid chromatography—tandem mass spectrometry. Baseline vitamin D proved deficient in over 80% of the subjects. The administration of magnesium resulted in significantly increased vitamin D levels in the intervention group versus the controls (*p* < 0.05). Magnesium supplementation improved vitamin D status in the studied postmenopausal women.

## 1. Introduction

Menopause is characterized by physiological changes with important variations in hormone levels. If left unchecked, this situation can lead to disease [1], including an increased risk of different types of cancer, cardiovascular disorders, osteoporosis and type 2 diabetes, among other conditions [2,3]. During this stage of life, women may experience weight gain and a redistribution of fat mass. Added to the hormonal alteration, this could adversely affect the status of different key micronutrients such as magnesium (Mg) and vitamin D in this population [4,5].

Magnesium is necessary for most reactions in the human body, and is a cofactor of more than 300 enzymes [6]. Mg is essential for the functioning of parathyroid hormone (PTH) and vitamin D. Hypomagnesemia during postmenopause needs to be monitored together with the status of those minerals closely related to phosphorus-calcium metabolism, in order to optimize homeostatic equilibrium and bone health. Magnesium supplementation may offer benefits in this regard [7,8]. The inclusion of Mg supplementation in postmenopausal women in the event of deficiency has been suggested by a number of authors, as it seems to improve postmenopausal symptoms, avoiding long-term systemic consequences [9,10,11]. In recent years, the interest in vitamin D has increased, due to the high prevalence of vitamin D deficiency worldwide [12]. Vitamin D plays a key role in phosphorus-calcium metabolism, improving the intestinal absorption of Ca, and regulating bone mineralization and the renal excretion of Ca [13,14]. To prevent poor vitamin D status, the monitoring of risk populations such as postmenopausal women is recommended with a view to preserving bone health [15]. However, new studies have also addressed the role of vitamin D in non-skeletal diseases [16,17].

Nowadays, the routine analytical determination of vitamin D is recommended in healthy risk groups such as postmenopausal women. However, such determinations are characterized by variability of the results obtained—thus suggesting the need to standardize the laboratory test protocol employed [18]. One of the methods currently used to measure vitamin D is enzyme immunoassay (EIA), which is the most widely used method in hospitals [19,20]. Use is currently also made of chromatography, which yields stable and reproducible results, and distinguishes between 25-OH-D_3_ and 25-OH-D_2_ [12]. In this respect, Liquid Chromatography—Tandem Mass Spectrometry (LC-MS/MS) is regarded as the gold standard, offering greater sensitivity, flexibility and specificity [18]. Unfortunately, LC-MS/MS cannot always be used, due to its high cost [20,21]. The technique of choice is therefore conditioned by the availability of resources [22,23]. In general, all techniques measure mainly 25-hydroxy-vitamin D (25-OH-D), because of its long half-life (one month) in plasma. Plasma vitamin D concentrations are conditioned not only by homeostatic regulation but also by lifestyle, environmental and sociocultural factors such as the use of sunscreens, the female gender, postmenopausal status and fat mass [24,25,26,27,28]. During postmenopause, vitamin D supplementation could be recommended in women with confirmed vitamin D deficiency, since it seems to be associated with an increase in bone mineral density and could improve future quality of life [29].

Therefore, the postmenopausal period could be associated with a genuine risk of deficiency of various minerals and vitamins, particularly Mg and vitamin D [30,31]. The present study was carried out to assess vitamin D status in a population of postmenopausal women in the province of Granada (Spain), with evaluation of the influence of a magnesium intervention.

## 2. Materials and Methods

### 2.1. Study Design and Intervention

This is an eight-week, double-blinded, placebo-controlled, randomized intervention trial (Figure 1). Participants were randomly assigned to one of two treatment groups: Placebo group (PG: 25 women) and Magnesium Group—500 mg/day of Mg (MG: 27 women). Randomization was performed in a 1:1 ratio using a table of random numbers, prepared by a researcher who did not participate in the data collection. Allocation concealment was ensured, as the referred researcher did not release the randomization code until the participants were recruited into the trial after all baseline measurements were completed. Mg supplements were supplied by Botánica Nutrients SL, Seville, Spain (Number B91070797), following the period of eight weeks recommended. Placebo capsules were made of the same size and color as Mg supplements for identical appearance and taste. The intervention was carried out in winter from January 15th to March 15th. The study was registered at the US National Institutes of Health (ClinicalTrials.gov) NCT03672513.

### 2.2. Study Participants

Fifty-two healthy postmenopausal women volunteers from the province of Granada, Spain aged between 44 and 76 years were recruited once they had been informed about the protocol. Inclusion criteria were (i) to present postmenopausal status (with at least 12 months of amenorrhea), (ii) to present low status in Mg obtained in a previous biochemical assessment, (iii) not present any pathology that could affect their nutritional status, (iv) not to be subjected to hormone replacement therapy (HRT), (v) not to take vitamin and mineral supplements. Women were excluded if they were unwilling to accept the randomization procedure. Written informed consent was obtained from all patients taking into account the approval of the Ethics Committee and the Research Committee of the Centre. The present study was conducted according to the principles of the Declaration of Helsinki and the approval by the Ethics Committee of the University of Granada (149/CEIH/2016), in accordance with the International Conference on Harmonization/Good Clinical Practice Standards.

Eligible participants of this study were 121 participants. Of these, 39 menopausal women were excluded because 18 women did not meet the inclusion criteria and 21 women declined to participle in the study after the initial interview, and so, 82 menopausal women were enrolled in the study and randomly assigned to the two arms (Figure 2). Of the 41 postmenopausal women that were allocated to intervention in the PG, a total of 16 women withdrew the study due time and supplementation commitment. In reference to the 41 postmenopausal women allocated into MG, a total of 13 postmenopausal women withdrew due to time and supplementation commitment, illness severity or not giving any reason. Of the 28 women included in the data analysis, one woman was excluded from the analysis because of insufficient blood sample was collected. Thus, 25 women in PG and 27 women in MG were enrolled in the present study.

### 2.3. Randomization and Blinding

Women were randomly assigned (simple randomization) to study groups (parallel design). In order to ensure comparable distribution across the treatment arms, women were stratified to balance baseline covariates. Both study participants and investigators were blinded to the group allocation. Initial and follow-up visits for evaluating dietary intake, body composition, biochemical and hormonal determination and antioxidant status parameters were performed at baseline and after two months of intervention.

### 2.4. Sample Size

We performed sample size calculation for our primary aim of a randomized controlled trial based on the influence of a Mg supplementation on vitamin D status. The number of participants to be included in the study was calculated on the basis of the change in vitamin D status after Mg intervention. To the best of our knowledge there were no available information regarding group difference changes on vitamin D in Mg intervened postmenopausal women. Therefore, we assumed a difference of 2.63 ng/mL as clinically meaningful based on previous observations in our group (unpublished data). A total of 68 participants were needed to detect a mean group difference of 2.63 ng/mL and a standard deviation of 3.85 ng/mL in vitamin D with a power of 80% and an alpha of 0.05, and assuming a maximum loss of 20% of participants (*n* = 82).

### 2.5. Compliance Evaluation

Adherence/compliance to nutritional intervention was determined as the percentage of all of the supplement capsules ingested throughout the study period. In addition, subjects were asked to keep daily records about side effects or other problems related to the supplements. Moreover, biochemical and clinical-nutritional parameters were taken at baseline and follow-up to evaluate the safety of the product and to verify the adverse effects.

### 2.6. Data Collection

All recorded data were obtained through the use of manual questionnaires administered by the interviewer that reflected information on personal data, sociodemographic aspects, an adequate diagnosis of the postmenopausal situation, smoking habits and physical activity [30]. 

### 2.7. Body Composition Analysis

Anthropometric recorded data were height (SECA^®^ Model 274), waist circumference (SECA^®^ Model 201), and body composition by bioelectrical impedance (Tanita MC-980 Body Composition Analyzer MA Multifrequency Segmental, Barcelona, Spain). The analyzer complied with the applicable European standards (93/42EEC, 90/384EEC) for use in the medical industry. Participants were informed in advance of the required conditions prior to the measurement: no alcohol less than 24 h before the measurement, no vigorous exercise less than 12 h prior to the measurement, no food or drink less than three h prior to the measurement, and no urination immediately before the measurement. All measurements were taken simultaneously during the morning in fasting conditions. Weight and BMI measurements were calculated and the compartmental analysis measured fat mass, fat free mass and muscle mass. The following measurements were taken: age, height, weight, BMI and fat percentage.

### 2.8. Intake Rating

Dietary nutrient intake was assessed using a manual 72 h-recall [30], taking into account a holiday and two non-holidays days, both at baseline and follow-up, which was administered by the interviewer. Recall accuracy was recorded with a set of photographs of prepared foods and dishes that are frequently consumed in Spain. The food intake assessment was converted to both energy and nutrients, determining the adequacy of the macronutrient and micronutrient intake according to the Recommended Dietary Allowance (RDA) for the female Spanish population within the age range included in our study [32] using Nutriber^®^ software (1.1.5. version, Barcelona, Spain).

### 2.9. Sample Treatment

A blood extraction in the morning in fasting conditions was performed at baseline and follow-up, being centrifuged at 4 °C for 15 min at 3000 rpm to extract the plasma. Once the plasma was removed from the tube, it was frozen at −80 °C until the analytical determination of the different parameters. All samples were measured in one run, in the same assay batch and blinded quality control samples were included in the assay batches to assess laboratory error in the measurements.

#### Measurement of Biochemical Parameters

PTH and osteocalcin levels were measured using EIA by colorimetric method (ECLIA, Elecsys 2010 and Modular Analytics E170, Roche Diagnostics, Mannheim, Germany). Vitamin D levels were measured by LC-MS/MS (Acquity UHPLC System I-Class Waters, Milford, USA) [33]. The biochemical values of vitamin D obtained were classified according to the reference values of 25-OH-D in plasma, being sufficiency >30 ng/mL, insufficiency 20–30 ng/mL and deficiency <20 ng/mL for total vitamin D [33]. The remaining biochemical parameters such as glucose, urea, uric acid, creatinine, triglycerides, total cholesterol, high density lipoprotein (HDL) and low density lipoprotein (LDL) cholesterol, total proteins, transferrin, albumin, homocysteine, bilirubin and transaminases levels, were determined in the Analysis Unit at the Virgen de las Nieves Hospital, Granada (ECLIA, Elecsys 2010 and Modular Analytics E170, Roche Diagnostics, Mannheim, Germany).

### 2.10. Statistical Analysis

The statistical analysis was performed using the SPSS 22.0 Software for MAC (SPSS Inc. Chicago, IL, USA). Descriptive analysis has been used for data expression, indicating the results of the numerical variables such as arithmetical mean, standard deviation (X ± SD) and standard error of the mean (SEM), and the results of the categorical variables were expressed in frequencies (%). As a previous step to the execution of a parametric model or not, the hypothesis of normal distribution was accepted using the Kolmogorov-Smirnov test. For the comparative analysis based on categorical variables, chi square test was used. For the comparative analysis based on baseline and follow-up, the paired t-test for parametric samples was used. For the comparative analysis based on groups, the unpaired t-test for parametric samples was used. Correlation analyses and partial correlation coefficients were performed with Pearson test. A *p* value less than 0.05 was considered statistically significant.

## 3. Results

The mean levels of plasma and erythrocyte Mg were 1.85 ± 0.25 (1.70–2.60) and 4.03 ± 0.71 (4.20–6.70) respectively. Our results showed that 27% of the postmenopausal women were deficient in plasma Mg and 67% were deficient in erythrocyte Mg at the beginning of the study. Given the deficiency justified here, the study population was randomly supplemented with Mg. 

Table 1 shows the general characteristics of the study population by groups. In both study groups, body mass index (BMI) was above and energy intake was below the reference values. Regarding Ca, 16.7% and 10.7% of the postmenopausal women in PG and MG did not reach two-thirds of the RDA at baseline. After the intervention, 20.0% of the women in PG and 11.1% of those in MG did not reach two-thirds of the RDA referred to Ca intake. With regard to Mg intake, 41.7% and 46.4% of the women in PG and MG, respectively, were below two-thirds of the RDA at baseline. Nevertheless, after Mg supplementation, 30% of the postmenopausal women in PG and 100% of those in MG reached two-thirds of the RDA for Mg. In the case of vitamin D intake, our results showed that 87.5% and 81.5% of the postmenopausal women in PG and MG, respectively, were below two-thirds of the vitamin D recommendations. After Mg supplementation, these figures were 75% and 92.6%.

Table 2 shows the biochemical parameters by group. In both groups, total cholesterol was above its reference values, and prealbumin significantly decreased in PG (*p* < 0.05) after the intervention. When using the LC-MS/MS method, the 25-OH-D levels were seen to have increased significantly after the intervention comparing baseline and follow-up (*p* < 0.05), though the levels were still below the recommended values. However, although an increase in 25-OH-D_3_ and 25-OH-D_2_ levels was also seen intra (MG) and inter-groups, the results were not statistically significant.

Figure 3a shows the distribution of 25-OH-D in the study population at baseline and after Mg intervention. Lesser data dispersion of the 25-OH-D levels was obtained after Mg supplementation when compared with baseline. Figure 3b shows the percentage of postmenopausal women with different vitamin D statuses by group. We found 80.8% of the study population to initially have vitamin D deficiency as established by LC-MS/MS. After the Mg intervention, the percentage of women in MG lacking in vitamin D decreased by about 20%.

## 4. Discussion

Previous scientific evidence indicates that postmenopausal women are at risk of suffering numerous micronutrient deficiencies. However, although vitamin D and Mg could be candidates for deficiency in this population, there is currently not enough evidence of the interaction between them. On the other hand, a large part of our study population was deficient in vitamin D as evidenced by LC-MS/MS and the vitamin D status was seen to improve in MG after the Mg intervention. 

Authors such as Rosanoff et al. [34] have affirmed that western populations (including Spain) are characterized by a low intake of Mg, since the latter is a predominant mineral in vegetables, and current consumption trends are towards an increased intake of animal products. Our results evidenced a pattern of low Mg consumption below the RDAs, with the exception of the MG population following the Mg intervention. In our study, Ca intake was seen to be within the recommended range for postmenopausal women. This is in contradiction to the findings of another study conducted in Spanish postmenopausal women, in which Ca intake fell short of the RDA. However, the data coincided with our own observation of vitamin D intakes below 50% of the RDA [35]. Another study involving a sample of 144 African women, in which dairy product consumption was lower, found that over 90% of the menopausal women analyzed failed to reach the RDA for Ca [36]. On the other hand, it should be noted that vitamin D intake in menopause is very low, as evidenced by studies such as that published by Rizzoli et al. [37], in which vitamin D intake among most postmenopausal women was seen to be very low in nine European countries. This is consistent with our own study, where most of the women failed to reach the RDA corresponding to vitamin D. 

In the present study, Mg intake showed a significant correlation (*r* = 0.451; *p* = 0.03) to the 25-OH-D levels. Authors such as Deng et al. [38] have argued that Mg intake is inversely proportional to 25-OH-D deficiency, independently of whether Mg is administered alone or in combination with vitamin D. This association was suggested to be due to the close relationship between Mg and vitamin D metabolism. Moreover, in our population, Mg and Ca intake were correlated (*r* = 0.689; *p* = 0.001). Authors such as Olza et al. [39] have mentioned the fact that Mg and Ca intake is based fundamentally on two food groups, namely cereals and dairy products, which are among the most widely consumed products in the Spanish population. Other authors such as Al-Musharaf et al. [40] have found vitamin D status to improve with the intake of Ca, as many food rich in Ca also have high a content of vitamin D. Nevertheless, we found no significant association between them, as well as no correlation to vitamin D intake, as this was too low—presenting high deficiency levels in almost all the women analyzed. These results could be explained by the data from studies such as that of Harris et al. [41], indicating that such correlations with vitamin D intake cannot be made, since its contribution depends on other elements such as genetic factors, as well as on exposure to the sun. 

Vitamin D levels are deficient in a large percentage of the population (Figure 3), and this pattern is observed in all parts of the world and at any latitude [42]. In our study, a large percentage of the population presented 25-OH-D levels below the reference values when analyzed by LC-MS/MS. Authors such as Park et al. [43] analyzed 25-OH-D levels by LC-MS/MS in a population of postmenopausal women, and recorded the same high prevalence of vitamin D deficiency (82%) as in our study. This confirmed that despite use of the gold standard for the analytical determination of vitamin D, the levels of this vitamin were low. Schmitt et al. [44] studied 25-OH-levels by EIA among 463 postmenopausal women and found only 32% of them to have sufficient levels. Vitamin D deficiency therefore is a generalized finding in the postmenopausal population, with high deficiency levels being reported by both chromatographic and immunological laboratory test methods.

Several authors [38,45,46,47] have reported Mg to exert synergic action with vitamin D, placing special emphasis on the effect of Mg upon the vitamin D binding protein (DBP), as well as on the enzymes that mediate in the hydroxylation of vitamin D in the liver and kidney. Thus, a high intake of either dietary or supplemented Mg could lessen the risk of vitamin D deficiency. According to our results, there was a significant increase in 25-OH-D levels after Mg supplementation (*p* < 0.05) when analyzed by LC-MS/MS. However, authors such as Melamed et al. [48] have pointed out that the administration of Mg supplements does not increase the 25-OH-D levels, despite the fact that Mg has a direct relationship with vitamin D metabolism. This could be explained by considering that if a population has very low vitamin D levels, the supplemented Mg might not be able to mediate the hydroxylation of enough vitamin D to improve vitamin D status. Hence, Mg supplementation is usually recommended together with vitamin D, advising Mg in greater proportion than vitamin D, in order to prevent all the Mg from being depleted by the hydroxylation of vitamin D [49].

On considering the results referred to vitamin D obtained with LC-MS/MS when comparing baseline with follow-up, different values were observed (Figure 3) according to the level of deficiency with respect to the reference values. The LC-MS/MS technique provided data indicating higher vitamin D deficiency at baseline compared with the data obtained (*p* < 0.05) in MG after the intervention. Granado-Lorencio et al. [50] studied the 25-OH-D levels of 32,363 general population samples using the EIA method, and suggested that the results obtained were unable to predict vitamin D deficiency, since the technique usually underestimates vitamin status, and even more so when 25-OH-D is present in low amounts. Other authors such as Klapkova et al. [51] consider that different methods other than LC-MS/MS likewise underestimate vitamin D status, thus affecting clinical decision making. Nikooyeb et al. [26] analyzed 275 general population serum samples using different methods, and argued that although the chromatographic techniques are the gold standard for the laboratory test determination of vitamin D, the results are comparable, since there are no major differences among the techniques. However, other authors such as Garg et al. [52] stated that although the differences in results obtained by the various vitamin D analytical methods have been reduced in recent years, it is advisable to use chromatographic techniques until full harmonization of the analytical methods for vitamin D is achieved. To date, the immunochemical methods have not been able to match the precision and specificity of the chromatographic techniques [53]. Accordingly, LC-MS/MS would be a more appropriate method in this scenario, since the results exhibit less dispersion, and do not usually underestimate the values, in coincidence with Atef et al., [12] who found the LC-MS/MS technique to estimate within normality ranges in the studied adult subjects. 

In addition to described findings, the present study has some strengths and limitations. As strength, the study is a randomized, placebo-controlled study in which nutritional intake of energy, macronutrients and related Mg and vitamin D minerals were controlled at baseline and follow-up. In this regard, we found that nutritional intake and high compliance to supplementation remained stable during the nutritional intervention. The present study used LC-MS/MS which is the gold standard analytical method, offering greater sensitivity, flexibility and specificity [18]. Despite it, this study has some limitations that should be considered as the small sample size. Although initially 82 women were randomly assigned to be supplemented, a total of 52 postmenopausal women completed the study (Figure 2). Although the primary outcome of the trial was to assess the influence of a Mg diet strategy on vitamin D status in postmenopausal women, the sample size in each group would allow us to preliminarily obtain significant results, although the results should be carefully considered. Clinical trials of the same nature and a similar sample size have shown a positive effect of different interventions on some parameters status in postmenopausal women [54,55,56]. Likewise, the sample size limitation did not allow us to make a more complex statistical approach since we did not have enough power to perform multivariate analyses and to be able to adjust our model based on possible confounding variables such as previously described age or BMI. 

## 5. Conclusions

Magnesium supplementation in the postmenopausal women of our study had a significant positive impact upon their vitamin D status. Most of the postmenopausal population presented inadequate plasma 25-OH-D levels. Future studies are needed to shed light on the vitamin D status of this risk population and to define protocols and strategies such as Mg intervention in postmenopausal women, with a view to improving their health and quality of life.

## Figures and Tables

**Figure 1 nutrients-12-02283-f001:**
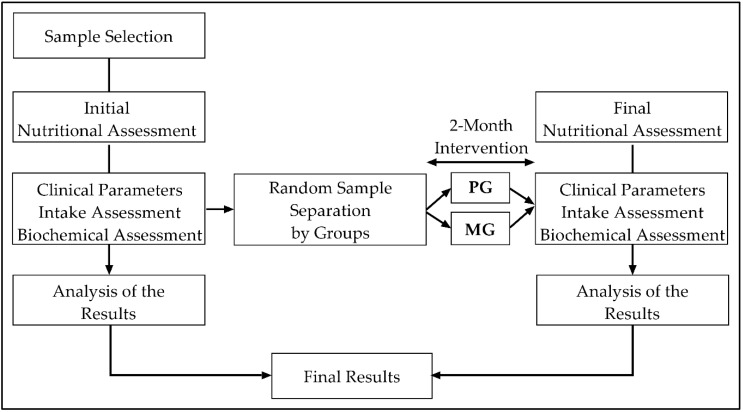
Study design. MG = Magnesium Group. PG = Placebo Group.

**Figure 2 nutrients-12-02283-f002:**
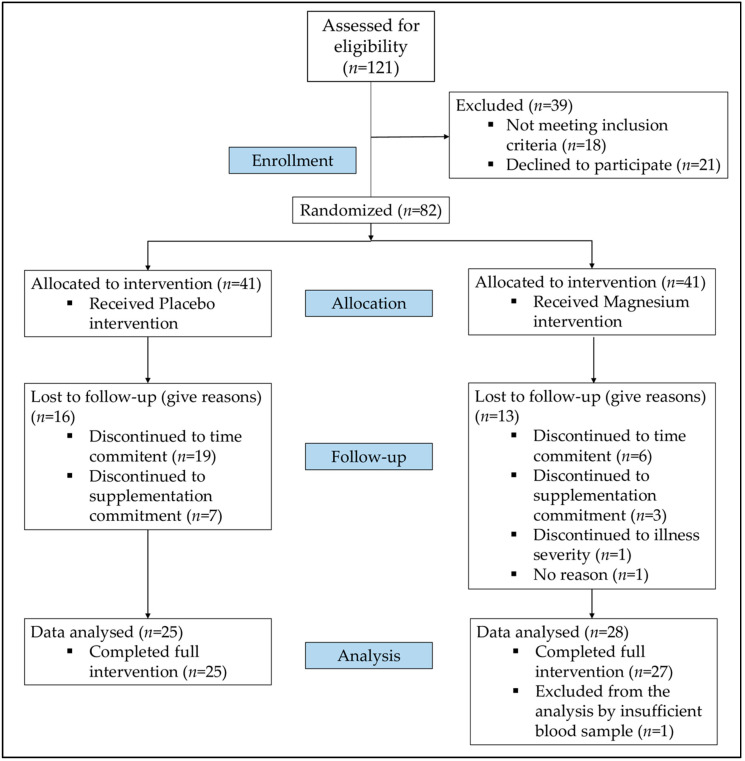
Flowchart of participants recruited, enrolled and involved in the clinical study.

**Figure 3 nutrients-12-02283-f003:**
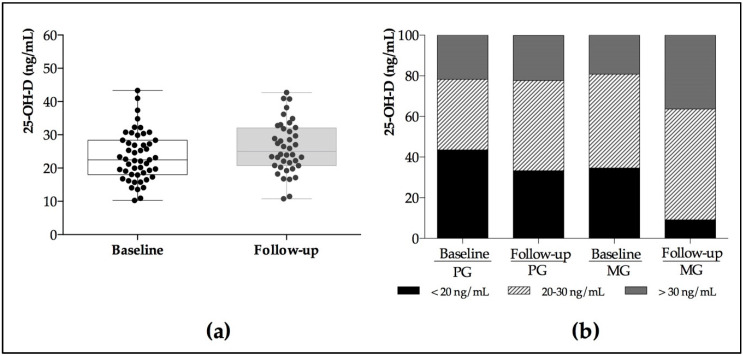
(**a**) Data distribution of 25-OH-D levels at baseline and follow-up. (**b**) Vitamin D status in the study population by groups. PG = Placebo Group. MG = Magnesium Group.

**Table 1 nutrients-12-02283-t001:** General characteristics of the study population by group.

Features	Reference Values	PG (*n* = 25)		MG (*n* = 27)		*p* Value PG Follow-Up	*p* Value MG Follow-Up	*p* Value Inter-Groups
		Baseline (Mean ± SD)	Follow-Up (Mean ± SD)	Baseline (Mean ± SD)	Follow-Up (Mean ± SD)			
Age (Years)	-	59.7 ± 9.15	59.7 ± 9.15	57.7 ± 7.58	57.7 ± 7.58	-	-	-
Weight (Kg)	-	69.1 ± 11.0	67.8 ± 11.2	69.3 ± 14.2	69.7 ± 13.6	0.23	0.43	0.62
Height (cm)	-	157.2 ± 6.01	157.2 ± 6.01	160.2 ± 6.11	160.2 ± 6.11	-	-	-
BMI (Kg/m^2^)	22.0–27.0	28.0 ± 4.30	27.7 ± 4.41	26.9 ± 4.88	27.7 ± 4.41	0.21	0.49	0.65
Blood pressure *n* (%)								
Normal blood pressure	-	10 (40)	-	14 (51)	-	-	-	-
High blood pressure	-	15 (60)	-	13 (49)	-	-	-	-
Physical exercise *n* (%)								
Sedentary	-	9 (36)	-	6 (23)	-	-	-	-
Non-sedentary	-	16 (64)	-	21 (77)	-	-	-	-
Smoking habit *n* (%)								
Non-smoker	-	18 (75)	-	23 (82)	-	-	-	-
Smoker	-	6 (25)	-	5 (18)	-	-	-	-
Educational level *n* (%)								
Basic educational level	-	10 (40)	-	11 (40)	-	-	-	-
Secondary or high educational level	-	15 (60)	-	16 (60)	-	-	-	-
Energy intake (Kcal)	2000.0	1339.5 ± 283.1	1232.9 ± 285.5	1307.1 ± 323.3	1323.4 ± 264.5	0.17	0.66	0.27
CHO intake (g/day)	275.0	146.6 ± 40.3	146.5 ± 33.4	149.5 ± 48.2	150.5 ± 48.5	0.84	0.89	0.75
Protein intake (g/day)	50.0	59.7 ± 14.2	57.1 ± 10.4	61.1 ± 17.6	63.4 ± 15.6	0.47	0.42	0.12
Fat intake (g/day)	70.0	56.1 ± 17.2	47.9 ± 18.5	53.2 ± 14.7	51.8 ± 12.7	0.15	0.66	0.41
Cholesterol intake (mg/day)	<300.0	150.6 ± 61.6	151.7 ± 64.4	154.1 ± 64.6	158.3 ± 76.8	0.87	0.81	0.75
Fiber intake (g/day)	>25.0	17.1 ± 10.6	16.2 ± 4.07	15.7 ± 7.72	16.5 ± 7.50	0.42	0.53	0.88
P intake (mg/day)	800.0	996.1 ± 257.7	993.9 ± 219.1	1002.6 ± 318.9	1038.8 ± 282.5	0.86	0.57	0.55
Ca intake (mg/day)	800.0–1000.0	728.2 ± 223.1	679.9 ± 168.6	873.5 ± 250.2	832.5 ± 210.6	0.69	0.56	0.01
Mg intake (mg/day)	320.0	237.4 ± 87.0	232.7 ± 55.8	219.3 ± 71.1	726.4 ± 59.9	0.99	0.001	0.001
Vitamin D intake (µg/day)	10.0	3.36 ± 3.00	4.34 ± 2.91	3.89 ± 3.62	3.62 ± 2.57	0.35	0.72	0.57

*n* = 52. BMI = Body Mass Index. CHO = Carbohydrates. P = Phosphorous. Ca = Calcium. Mg = Magnesium. Baseline and follow-up values are expressed as mean ± standard deviation. Both for intra-group and inter-groups *p*-value, paired and unpaired t-student test was used. Categorial variables are expressed as continuous as sample size (*n*) and percentage of subjects (%), and chi-square test was used. PG = Placebo Group. MG = Magnesium Group.

**Table 2 nutrients-12-02283-t002:** Biochemical parameters of the study population by group.

Features	Reference Values	PG (*n* = 25)		MG (*n* = 27)		*p* Value PG Follow-Up	*p* Value MG Follow-Up	*p* Value Inter-Groups
		Baseline (Mean ± SD)	Follow-Up ((Mean ± SD)	Baseline ((Mean ± SD)	Follow-Up ((Mean ± SD)			
Glucose (mg/dL)	70.0–110.0	96.0 ± 19.8	95.8 ± 19.9	90.1 ± 11.1	95.8 ± 18.9	0.63	0.25	0.047
Transferrin (mg/dL)	200.0–360.0	285.9 ± 39.7	272.8 ± 43.4	284.2 ± 47.7	272.8 ± 43.4	0.89	0.18	0.71
Prealbumin (mg/dL)	20.0–40.0	26.6 ± 5.03	24.7 ± 5.01	24.8 ± 6.45	24.7 ± 5.01	0.001	0.06	0.62
Albumin (mg/dL)	3.50–5.20	4.50 ± 0.20	4.45 ± 0.27	4.50 ± 0.21	4.45 ± 0.27	0.62	0.055	0.57
Homocysteine (µmol/L)	<13.0	12.5 ± 6.45	12.7 ± 4.78	11.5 ± 4.25	12.7 ± 4.78	0.27	0.66	0.46
Creatinine (mg/dl)	0.50–0.90	0.75 ± 0.16	0.75 ± 0.16	0.68 ± 0.11	0.75 ± 0.16	0.31	0.81	0.08
Total bilirubin (mg/dL)	0.10–1.20	0.49 ± 0.13	0.55 ± 0.21	0.47 ± 0.11	0.55 ± 0.21	0.13	0.95	0.26
LDH (U/L)	110.0–295.0	182.8 ± 29.3	181.1 ± 26.1	192.5 ± 26.7	181.1 ± 26.1	0.96	0.41	0.81
Urea (mg/dL)	10.0–50.0	36.2 ± 10.2	36.7 ± 9.35	34.3 ± 8.98	36.7 ± 9.35	0.87	0.65	0.44
Uric acid (mg/dL)	2.40–5.70	4.51 ± 0.98	4.70 ± 1.02	4.43 ± 1.23	4.70 ± 1.02	0.21	0.26	0.21
Triglycerides (mg/dL)	50.0–200.0	115.8 ± 68.9	112.3 ± 62.2	111.1 ± 50.6	112.3 ± 62.2	0.79	0.41	0.80
HDL (mg/dL)	40.0–60.0	62.6 ± 11.2	64.1 ± 12.3	66.6 ± 14.4	64.1 ± 12.3	0.04	0.06	0.95
LDL (mg/dL)	70.0–190.0	134.5 ± 35.3	137.6 ± 30.5	130.4 ± 26.4	137.6 ± 30.5	0.76	0.06	0.19
Total cholesterol (mg/dL)	110.0–200.0	224.1 ± 39.7	221.4 ± 31.4	224.7 ± 30.1	221.4 ± 31.4	0.86	0.09	0.64
Osteocalcin (ng/mL)	15.0–46.0	17.4 ± 9.45	18.1 ± 7.21	16.8 ± 10.4	18.1 ± 7.21	0.69	0.22	0.91
PTH (pg/mL)	20.0–70.0	50.7 ± 15.8	53.3 ± 34.4	52.9 ± 17.2	53.3 ± 34.4	0.46	0.07	0.29
Ca (mg/dL)	8.60–10.2	9.31 ± 0.31	9.14 ± 0.44	9.27 ± 0.51	9.13 ± 0.49	0.31	0.07	0.98
P (mg/dL)	2.70–4.50	3.45 ± 0.45	3.57 ± 0.55	3.42 ± 0.53	3.60 ± 0.49	0.49	0.07	0.88
25–OH–D LC–MS/MS (ng/mL)	30.0–100.0	23.0 ± 8.99	24.2 ± 7.71	23.6 ± 5.70	27.8 ± 7.56	0.81	0.049	0.14
25–OH–D_3_ LC–MS/MS (ng/mL)	>20	18.0 ± 8.37	19.7 ± 8.00	17.7 ± 6.25	21.1 ± 7.40	0.52	0.13	0.57
25–OH–D_2_ LC–MS/MS (ng/mL)	>10	4.99 ± 2.11	4.55 ± 2.74	5.86 ± 3.05	6.80 ± 7.16	0.31	0.41	0.22

*n* = 52. LDH = Lactate dehydrogenase. HDL = High density lipoprotein. LDL = Low density lipoprotein. PTH = Parathyroid hormone. Ca = Calcium. P = Phosphorous. LC-MS/MS = Liquid chromatography—tandem mass spectrometry. Baseline and follow-up values are expressed as mean ± standard deviation. For intra-group and inter-groups *p*-value, paired and unpaired t-student test were used respectively. PG = Placebo Group. MG = Magnesium Group.

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
