# Peer review of "Response of Vitamin D after Magnesium Intervention in a Postmenopausal Population from the Province of Granada, Spain"

_nutrients, 2020, doi:10.3390/nu12082283_

Round 1

Reviewer 1 Report

The authors have removed the socioeconomic variable from this revision after it being pointed out that they had incorrectly treated categorical variables as continuous variables in the first revision. These should be returned to the manuscript (appropriately expressed as categorical variables), as they are necessary for demonstrating the success of randomisation and are especially critical here as the study had a large number of drop outs, and so it will be important to show that there are no differences in the demographics due to this. 

The authors have not adequately addressed the small sample size in their study, simply saying in the discussion "it still got significant results" is not enough. A small study can still get spuriously significant results. A proper discussion of the sample size is required. 

You have also said p LES THAN 0.05 is siginificant, but it appears that the "significant" p=0.05, this is not technically less than 0.05 and this should be addressed. 

Total vit D is significantly different (barely!) after treatment but D2 and D3 are not, this needs addressing.    

Author Response

Response to Reviewer 1 Comments

Comments and Suggestions for Authors

Point 1:

The authors have removed the socioeconomic variable from this revision after it being pointed out that they had incorrectly treated categorical variables as continuous variables in the first revision. These should be returned to the manuscript (appropriately expressed as categorical variables), as they are necessary for demonstrating the success of randomization and are especially critical here as the study had a large number of drop outs, and so it will be important to show that there are no differences in the demographics due to this. 

Response 1:

The sociodemographic variables were again considered and included in table 1 in a qualitative view. Therefore, the chi-square analysis was performed to verify any differences in the sample distribution according to each of the sociodemographic variables studied. The statistical method used has been descripted in materials and methods in statistical analysis.

Point 2:

The authors have not adequately addressed the small sample size in their study, simply saying in the discussion "it still got significant results" is not enough. A small study can still get spuriously significant results. A proper discussion of the sample size is required. 

Response 2:

A new section called sample size has been added to the methods section. We thank the reviewer for his suggestion. Likewise, in the limitations section, we have also justified that several clinical trials whose purpose was to determine changes in vitamin D status against different interventions, have shown positive results with sample sizes similar to those of the present study.

Point 3:

You have also said p LES THAN 0.05 is significant, but it appears that the "significant" p=0.05, this is not technically less than 0.05 and this should be addressed. 

Response 3:

Dear reviewer, there was a typographical error and the value was rounded from p = 0.049 to p = 0.05. This error has been modified in table 2 of the results.

Point 4:

Total vit D is significantly different (barely!) after treatment but D2 and D3 are not, this needs addressing.   

Response 4:

As the reviewer suggests, we have addressed the non-statistically significant increase in 25-OH-D2 and 25-OH-D3 levels. As far as known, the variance is a statistic that substantially affects the statistical significance. The relationship is direct, i.e. the greater the dispersion, the greater the probability of rejecting the null hypothesis since lower values of the probability associated with the statistician are obtained (Kline, 2005). Given that the variance of a composite variable is no more than the sum of the variances of those variables that make it up, the dispersion of the composite variable is greater than the dispersion of the individual variables. In this case, even though the variables are added together (and therefore their variances) the sample size remains constant. Therefore, it is not surprising that the result of the composite variable is significant (p=.049) and the effect of the simple variables is not (p=.13 and p=.41 respectively).

- Kline, R.B. (2005). Beyond significance testing. Reforming data analysis methods in behavioral research. Washington: APA books

Reviewer 2 Report

Although the authors have responded to all points raised, two major points of criticism persist. The study design is still problematic. Including such a small number of postmenopausal women seems rather arbitrary. There is no good rationale to excclude other age-groups and males from such a study. In addition, the manuscript is still too long.

Author Response

Response to Reviewer 2 Comments

Comments and Suggestions for Authors

Point 1:

Although the authors have responded to all points raised, two major points of criticism persist. The study design is still problematic. Including such a small number of postmenopausal women seems rather arbitrary. There is no good rationale to exclude other age-groups and males from such a study.

Response 1: 

A new section called sample size has been added to the methods section. We thank the reviewer for his suggestion. Likewise, in the limitations section, we have also justified that several clinical trials whose purpose was to determine changes in vitamin D status against different interventions, have shown positive results with sample sizes similar to those of the present study.

Point 2:

In addition, the manuscript is still too long.

Response 2:

We thank the reviewer for his suggestion. Therefore, information that is not very relevant has been removed from the introduction and materials and methods section in order to shorten the manuscript.

Point 3:

There is no good rationale to exclude other age-groups and males from such a study.

Response 3:

With regard to the reviewer's comment, we appreciate the suggestion made. However, we differ on the inclusion of the male population. As indicated in this paper, the study group is postmenopausal women. This group presents a series of characteristics resulting from the disappearance of the menstrual period and a hormonal alteration widely described in the literature. Considering this, in men it would not be possible to assess these changes because both at a biological and physiological level, the hormonal response is different from women.

On the other hand, the inclusion of different age ranges would have made possible the inclusion of a more heterogeneous sample. In our study, we did not stratify by age because we considered that our sample size was not large enough to have an effect resulting from this stratification. If we had stratified by age, we would have probably lost any significant effect because the groups created would have been too small. We thank the reviewer again for this point, because the continuity of our research in our postmenopausal sample will make us consider such wise suggestions.

Round 2

Reviewer 1 Report

N/A